# A Novel High-Q Dual-Mass MEMS Tuning Fork Gyroscope Based on 3D Wafer-Level Packaging

**DOI:** 10.3390/s21196428

**Published:** 2021-09-26

**Authors:** Pengfei Xu, Chaowei Si, Yurong He, Zhenyu Wei, Lu Jia, Guowei Han, Jin Ning, Fuhua Yang

**Affiliations:** 1Engineering Research Center for Semiconductor Integrated Technology, Institute of Semiconductors, Chinese Academy of Sciences, Beijing 100083, China; xupengfei@semi.ac.cn (P.X.); schw@semi.ac.cn (C.S.); hyr617@semi.ac.cn (Y.H.); zywei97@semi.ac.cn (Z.W.); jialu@semi.ac.cn (L.J.); hangw1984@semi.ac.cn (G.H.); ningjin@semi.ac.cn (J.N.); 2College of Materials Science and Opto-Electronic Technology, University of Chinese Academy of Sciences, Beijing 100049, China; 3School of Electronic, Electrical and Communication Engineering, University of Chinese Academy of Sciences, Beijing 100049, China; 4State Key Laboratory of Transducer Technology, Chinese Academy of Sciences, Beijing 100083, China

**Keywords:** tuning fork gyroscope, MEMS, 3D packaging, high Q-factors

## Abstract

Tuning fork gyroscopes (TFGs) are promising for potential high-precision applications. This work proposes and experimentally demonstrates a novel high-Q dual-mass tuning fork microelectromechanical system (MEMS) gyroscope utilizing three-dimensional (3D) packaging techniques. Except for two symmetrically decoupled proof masses (PM) with synchronization structures, a symmetrically decoupled lever structure is designed to force the antiparallel, antiphase drive mode motion and eliminate low frequency spurious modes. Thermoelastic damping (TED) and anchor loss are greatly reduced by the linearly coupled, momentum- and torque-balanced antiphase sense mode. Moreover, a novel 3D packaging technique is used to realize high Q-factors. A composite substrate encapsulation cap, fabricated by through-silicon-via (TSV) and glass-in-silicon (GIS) reflow processes, is anodically bonded to the wafer-scale sensing structures. A self-developed control circuit is adopted to realize loop control and characterize gyroscope performances. It is shown that a high-reliability electrical connection, together with a high air impermeability package, can be fulfilled with this 3D packaging technique. Furthermore, the Q-factors of the drive and sense modes reach up to 51,947 and 49,249, respectively. This TFG realizes a wide measurement range of ±1800 °/s and a high resolution of 0.1°/s with a scale factor nonlinearity of 720 ppm after automatic mode matching. In addition, long-term zero-rate output (ZRO) drift can be effectively suppressed by temperature compensation, inducing a small angle random walk (ARW) of 0.923°/√h and a low bias instability (BI) of 9.270°/h.

## 1. Introduction

With the development of designing and manufacturing techniques, microelectromechanical system (MEMS) gyroscopes have been widely adopted in the fields of consumer and industrial applications for their low cost, facile integration and low power consumption [1,2,3]. Tuning fork gyroscopes (TFGs), composed of a decoupled geometry framework and two identical single-mass structures, have been attracting much attention due to their inherent common-mode suppression ability, high vibration immunity near the operating frequency and insensitivity to external accelerations [4,5]. TFGs operate based on Coriolis coupling between two orthogonal modes (the drive mode and the sense mode) [6]. In addition, they have two resonance frequencies corresponding to the in-phase and the antiphase modes in drive mode [7]. The in-phase vibration mode is susceptible to the vibration of a linear environment. It even impacts the vibration response of the antiphase vibration mode [8], which is usually eliminated with intricate device structures in practical applications. Nevertheless, the antiphase drive mode can eliminate low-frequency spurious modes and realize common-mode suppression [9]. Hence, the antiphase drive mode is usually utilized as the dominant mode of TFGs.

Various system-level noise sources, such as mechanical Brownian noise and flicker noise, are the main factors that affect TFG performances [10]. Enhancement of the quality factors (Q-factors) is key to boosting performance, putting forward new requirements for the design and package of gyroscopes [11]. By raising Q-factors of drive and sense modes, the performances of TFGs, such as power consumption, resolution and bias instability (BI), can be greatly promoted. The sensitive and fragile structure can be shielded from external impacts with a suitable package structure, and high Q-factors are thus obtained. On this occasion, a three-dimensional (3D) wafer-level packaging technology, combined with anodic bonding and vertical interconnection techniques, has a great potential in realizing smaller size TFGs with high Q-factors and superior hermetic packages [12,13,14].

In this work, a novel high-Q dual-mass MEMS TFG with 3D wafer-level packaging is reported. Compared with conventional TFGs, a symmetrically decoupled lever structure is designed to force the antiparallel, antiphase drive-mode motion, ensuring the suppression of common modes and the parasitical low-frequency in-phase mode. A linearly coupled antiphase sense mode with momentum and torque balances is adopted to reduce energy dissipation through the substrate, and the mechanical sensitivity and precision are greatly improved. Furthermore, a novel 3D packaging technique is used to realize a stable encapsulation with high Q-factors [15]. The sensing structures are entirely fabricated on silicon instead of a silicon-on-insulator platform to reduce the CET mismatch between three structure layers. Moreover, a composite substrate encapsulation cap, fabricated with through-silicon-via (TSV) and glass-in-silicon (GIS) reflow processes, is anodically bonded to the sensing structures at wafer scales. Embedded vertical silicon pillars with low resistivity are adopted to achieve high-reliability electrical connections. This 3D packaging technique realizes stable encapsulation with high Q-factors and simplifies the manufacturing process, and the air impermeability and reliability of TFGs are also drastically improved. The manufactured TFGs are tested and characterized with a self-developed control circuit system. The experimental results show that this TFG realizes ultra-high Q-factors, a wide measurement range and a small-scale factor of nonlinearity. Moreover, it achieves high sensitivity and precision under automatic mode-matching. The long-term zero-rate output (ZRO) drift can be effectively suppressed by temperature compensation, dramatically improving the angle random walk (ARW) and BI. This novel high-Q-factor TFG with 3D architecture creates new options for high-precision applications.

## 2. Design and Manufacture of Dual-Mass TFGs

A schematic illustration of the dual-mass TFG sensing element is shown in Figure 1. It is composed of two identical symmetrically decoupled proof masses (PMs), two symmetrically decoupled levers to force the antiphase drive mode motion and linearly coupled flexures towards the antiphase sense-mode motion. To study the mode motions of TFGs under electrostatic excitation, a field simulation for the designed structure is achieved with COMSOL Multiphysics software, and the results are shown in Figure 2. When an electrostatic excitation is applied to the drive elements, the two symmetrically decoupled levers force the proof masses into an antiparallel and antiphase drive-mode motion in the drive axis (*x*-axis), as can be seen in Figure 2a, ensuring the suppression of common modes and the elimination of parasitical low-frequency in-phase mode [9]. When the TFG rotates around the z-axis, the Coriolis acceleration is induced and the masses in sense mode move with an antiphase versus the substrate in the sense axis (*y*-axis), as shown in Figure 2b. A linearly coupled antiphase sense mode is adopted to reduce energy dissipation through the substrate based on momentum and torque balances by reducing the anchor loss and thermoelastic damping (TED). It thus greatly improves the mechanical sensitivity and precision of TFGs.

The manufacturing process flow of 3D wafer-level-packaged dual-mass TFGs is shown in Figure 3a, and the technical details are listed as follows:

(1)A 4-inch p-type <100>-oriented low-resistivity (0.0009 Ω⋅cm) silicon wafer, suffering a~400 μm-deep trench etch through deep reactive ion etch (DRIE), is used as a platform for the glass reflow process.(2)The silicon substrate and borosilicate glass wafer, which have the same thickness and an approximate coefficient of thermal expansion (CTE), are anodically bonded under 10^−5^ mbar at 330 °C. Then, the bonded wafer is annealed in a high-temperature furnace (1000 °C) for 2 h to ensure that the silicon grooves are fully filled with reflux glass under a large pressure difference.(3)After slow cooling, a double-sided chemical mechanical polishing (CMP) process is used to fabricate the vertical silicon feedthrough, and the composite substrate (fabricated with through-silicon-via (TSV) and glass-in-silicon (GIS) reflow processes) surface is then treated for the subsequent anodic bonding process.(4)A low-resistivity silicon wafer is used as the sensing layer, where a 10 μm-deep trench is etched out as the movable cavity of the sensing structure. The silicon wafer is then anodically bonded with the composite substrate and thinned to the same thickness of the composite substrate via single-sided CMP.(5)Lateral trenches are etched out with DRIE on the sensing structure layer using an oxide mask to realize silicon sensing structures and surrounding electrodes. The sensing structure layer surface is also treated for the anodic bonding process.(6)A borosilicate glass wafer masked by patterned CrAu (50 nm/100 nm) is etched to obtain the glass cavity by local wet etching (HF/HCl/DI.H_2_O = 10:1:30), and an 1 μm-thick highly porous Zr-based getter material is sputtered inside the glass cavity to empty active gases such as H_2_O, H_2_, CO_2_ and N_2_ [16]. This glass cap is then anodically bonded with the silicon sensing structures at a pressure of~10^−5^ mbar; then, the getter material is activated at 400 °C. Finally, a 1 μm-thick aluminum is deposited and then patterned by local wet etching on the surface of the composite substrate encapsulation cap to provide electrical connections.

The TFGs are composed of three structure layers: the glass cap, the sensing silicon structure and the composite substrate encapsulation cap, as depicted in Figure 3b. To simplify the manufacturing process and improve the reliability of 3D packaging, the embedded vertical low-resistivity silicon pillars are used to realize a high-reliability electrical connection. The silicon sensing structures are entirely fabricated on silicon instead of silicon-on-insulator substrates to reduce CET mismatch between these three layers. Moreover, the chamber vacuum is boosted by sputtering getter materials, and this 3D packaging technique is used to enhance the air impermeability and achieve stable packaging with ultra-high Q-factors.

A photograph of as-fabricated 3D wafer-level-packaged dual-mass TFGs is shown in Figure 4a. It can be seen that there is a mass of excellent gyroscopes on the whole 4-inch wafer, whose size is 4 mm × 5 mm. To further study the surface topography of internal MEMS structure, a scanning electron microscope (SEM) (NanoSEM650) is used to observe the detailed morphologies of the anchors, the masses, the cantilever beams, the capacitance combs and other structures. As shown in Figure 4b, the whole MEMS structure is flat, and no warping is found. There are no local structural or interlayer adhesions. Additionally, the surfaces, sidewalls and bottoms of MEMS structures are smooth and free of stain, which is essential to maintain good device characteristics. Furthermore, an atomic force microscopy (AFM) is used to characterize the surface roughness of the sensing silicon structure, where a tapping mode is adopted, and the scanning range is 1 μm × 1 μm. As illustrated in Figure 5a, the surface of the sensing silicon structure is smooth, and no particle pollution and obvious defects were found. The measured root mean square (RMS) roughness is only 0.187 nm according to the three-dimensional image shown in Figure 5b.

## 3. Characterization and Experimental Results

In order to further characterize the reliability of the 3D packaging technique and the performance of dual-mass TFGs, the TFGs are mounted on a special shell and then connected with a self-developed control circuit. As illustrated in Figure 6, this control circuit is mainly composed of: the front-end analog circuit, signal acquisition and conversion circuit and signal processing and loop control circuit. The front-end analog circuit, composed of ring diodes and amplifiers, is responsible for capacitance signal reading and conversion, signal amplification and filtering. Two 14-bits analog-to-digital converters (ADCs), three 14-bits digital-to-analog converters (DACs) at a sampling rate of 200 kHz are used for signal acquisition and conversion. Furthermore, a FPGA board (spartan-6, XC6SLX45 core board manufactured by broadon technology, Beijing, China) is used to realize signal processing and loop control, where the phase-locked loop (PLL) and automatic gain control (AGC) loops are adopted to stabilize gyroscope vibration amplitude and frequency. Additionally, an automatic mode-matching loop based on electrostatic force is used to eliminate the frequency splitting, improving the sensitivity, the precision and the scale factor of TFGs. Notably, the sense mode always operates in the equilibrium position and collects the angular output by utilizing a force-to-rebalance (FTR) closed-loop detection and quadrature error correction system [17].

To characterize the structural characteristics of the dual-mass TFGs, the frequency sweep properties (spanning 3.2 kHz) are detected using a dynamic signal analyzer (Keysight 35670A), and the results are shown in Figure 7. The resonance frequency of the antiphase drive mode is 12,824 Hz, which is 720 Hz-higher than that of the low-frequency in-phase drive mode (12,104 Hz), ensuring the suppression of common modes and the elimination of a parasitical low-frequency in-phase mode. The resonance frequency of the linearly coupled antiphase sense mode is 12,768 Hz, which is 56 Hz-lower than the master frequency of the drive-mode, as demonstrated in Figure 7a. An automatic mode-matching loop is proposed to reduce the frequency splitting between the two modes, and the resonance frequency of the antiphase sense mode is adjusted to 12,823.5 Hz with a negative electrostatic spring effect. It is noted that the resonance frequency difference is inevitable for the sake of a stable vibration. The resonance frequency of the sense mode suffering automatic mode-matching is shown in Figure 7b. The mechanical sensitivity and the signal-to-noise ratio (SNR) of the angular output are dramatically enhanced.

As mentioned above, a novel 3D packaging technique is used to realize high Q-factors’ encapsulation towards high-performance TFGs with low noise and high sensitivity. The fundamental mechanical Brownian noise of gyroscopes is related to the Q-factors [18], which can be expressed as follows:(1)$$\Omega_{z\left(Brownian\right)}\;\propto \sqrt{\frac{1}{Q_{Effect-Sense}}}$$
where $Q_{Effect-Sense}$ is the effective Q-factor of the sense mode. The thermo-mechanical noise can be reduced by increasing Q-factors. The Q-factor is defined as the ratio of the stored energy to the dissipated energy within a resonance cycle. According to the expression proposed in [19], the Q-factor can be calculated as follows:(2)$$Q\approx \;\pi \frac{1}{\lambda }f_{r}=\pi tf_{r}$$
where $f_{r}$ is the resonance frequency and t ($\frac{1}{\lambda }$) is the relaxation time, which can be acquired by studying the free vibrations in the time domain [15].

As shown in Figure 8, the vibration amplitude sharply decreases with time, and the variation trend follows an exponential form when the drive force is turned off. The exponential fitting results of the curve peaks are presented in the insets, from which relaxation time t can be obtained. The inherent resonance frequencies for the antiphase drive and sense mode are 12,824 Hz and 12,768 Hz (where the bias voltage $v_{\rho }$ = 15 V), respectively. The derived relaxation times are 1.2894 s and 1.2278 s for the drive and sense modes; hence, calculated drive quality factor $Q_{x}$ is 51,947, and sense quality factor $Q_{y}$ is 49,249, respectively. The high Q-factors of TFGs suggest that a reliable package with a high air impermeability can be effectively achieved with our 3D packaging technique.

The packaged gyroscope board is placed on an angular velocity table (TBL-S1101-AT03) to test its angular rate performance after automatic mode-matching. When an electrostatic excitation (DC = 15 V and AC = 2.4 Vrms) is applied to the drive elements, the proof masses work in the antiparallel, antiphase drive mode motion due to the symmetrically decoupled levers. Multiple angular rates are applied in the *z*-axis during the practical test. Averaged output data are calculated from multiple real-time data collected within 2 min for each angular input. The full-scale testing results, together with the nonlinearity, are shown in Figure 9. This dual-mass TFG has a high resolution of 0.1°/s, and the scale factor is 1.345 mV/(°/s), with a nonlinearity of 720 ppm in a ±1800°/s full-scale range. Experimental results show that the dual-mass TFG has a good responsivity and a high resolution for angular input. Moreover, it has a wide measurement range and high linearity. The key parameters of the dual-mass TFGs are shown in Table 1.

Room-temperature ZRO characteristics are measured with an FTR closed-loop detection and quadrature error correction system, in which the output angular rate data are recorded for 5 h at a sampling rate of 1 kHz. As shown in Figure 10, the ZRO results show an obvious drift tendency at the initial sampling stage. However, this drift gradually decreases with time and stabilizes at ~0.093°/s after a duration of ~25 min.

To excavate the inner mechanism on ZRO drift and further improve the output stability of TFGs, an elaborate study was carried out. The resonance frequencies and device temperatures, together with ZRO results, are synchronously measured and recorded at two modes during the whole sampling process. The drive-mode resonance frequency characteristics are shown in Figure 11a, where the frequency drift decreases with sampling time and stabilizes after a while. The total frequency drift is approximately 1.2 Hz during the 5 h-long sampling process. It can be inferred that there exists a similar variation tendency between drive-mode resonance frequency drifts and ZRO drifts. Furthermore, the real-time device temperatures detected by a temperature sensor module are given in Figure 11b for packaged TFGs. The device temperature varies with time at a similar tendency for drive-mode resonance frequency, which may be related to temperature-dependent characters of Young’s modulus for TFGs [20]; the total temperature variation is about 2.9 °C.

Given this, temperature compensation is accomplished for measured room-temperature ZRO data with the drive-mode resonance frequency as a reference value. A third-order temperature compensation method is adopted here to improve the fitting accuracy, and the fitted ZRO results are shown in Figure 12. It is revealed that a temperature compensation method can effectively suppress long-duration ZRO drifts on account of the strong correlations between ZRO, resonance frequency and temperature. As such, the performance of our dual-mass TFG can be greatly improved.

Allan standard variance analysis is performed for fitted ZRO results, and the analysis results for five typical sets of TFGs are shown in Figure 13. These curves are roughly overlapped, meaning a good within-wafer uniformity of our TFG fabrication processes. Two vital TFG parameters, ARW and BI, are extracted from each Allan standard variance plot and listed in Table 2 for every tested TFG. The ARWs are 0.947°/√h, 0.976°/√h, 0.891°/√h, 0.883°/√h and 0.918°/√h for these five TFGs (Sample 1~5), respectively. Correspondingly, their BIs are 9.277°/h, 9.699°/h, 9.175°/h,9.185°/h and 9.015°/h, respectively. The averaged ARW and BI values are only 0.923°/√h and 9.270°/h for the above five TFGs. The small ARW values and the small BI values testify that our dual-mass TFGs possess a small output fluctuation and a high precision, providing new options for future high-precision applications.

## 4. Conclusions

In conclusion, we present a novel high-Q dual-mass MEMS TFGs with a 3D packaging technique. Symmetrically decoupled lever structures are designed to force the antiparallel, antiphase drive-mode motion and ensure the suppression of common modes and the parasitical low-frequency in-phase mode. The energy dissipation through the substrate is reduced with the linearly coupled, momentum- and torque-balanced antiphase sense mode, greatly improving the mechanical sensitivity and precision of TFGs. To realize high Q-factors and air-impermeable encapsulation, a novel wafer-level 3D packaging technique is used towards high-reliability electrical connections, reduced CET mismatches and simplified fabrication processes. The drive-mode and sense-mode Q-factors are ultra-high (51,947 and 49,249, respectively) for this dual-mass MEMS TFG, and the measurement range reaches up to ±1800°/s with a scale factor nonlinearity of 720 ppm. It is shown that this TFG realizes a high resolution of 0.1°/s, and the long-duration ZRO drift can be effectively suppressed by a third-order temperature compensation method. Furthermore, a small ARW of 0.923°/√h and a low BI of 9.270°/h are observed, indicating that this dual-mass MEMS TFG has potential for future high-precision applications.

## Figures and Tables

**Figure 1 sensors-21-06428-f001:**
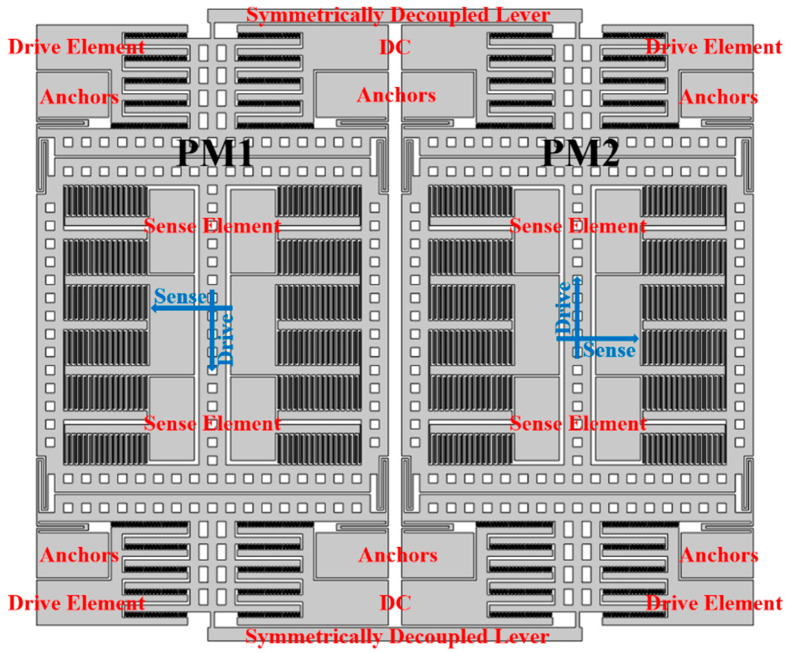
A schematic illustration of the dual−mass TFG sensing element.

**Figure 2 sensors-21-06428-f002:**
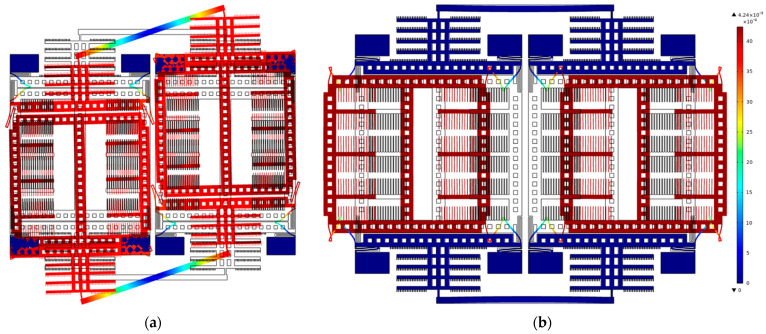
The simulated results of dual−mass TFGs with symmetrically decoupled lever structures: (**a**) the antiparallel, antiphase drive−mode motion, and (**b**) the linearly coupled antiphase sense−mode motion.

**Figure 3 sensors-21-06428-f003:**
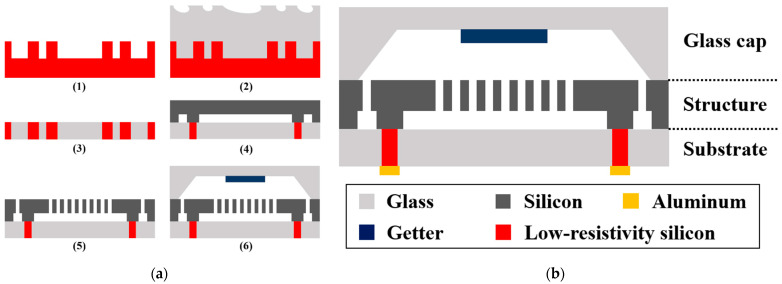
(**a**) The manufacturing process flow of 3D wafer-level-packaged dual-mass TFGs. (**b**) The device structure of the sensing element.

**Figure 4 sensors-21-06428-f004:**
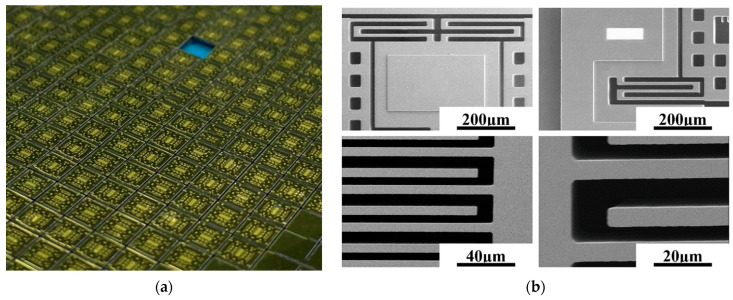
(**a**) A photograph of 3D wafer-level-packaged dual-mass TFGs; (**b**) SEM images for the MEMS structure.

**Figure 5 sensors-21-06428-f005:**
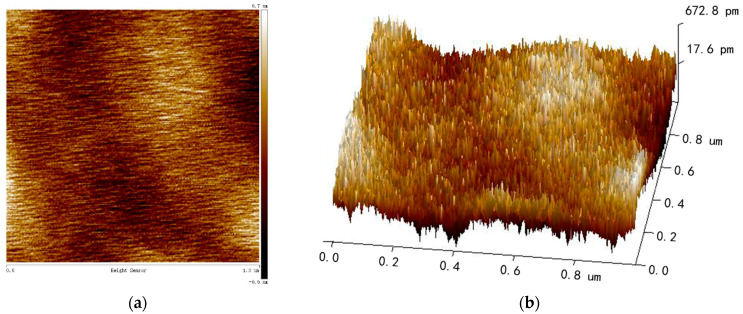
The roughness measurement results of (**a**) two- and (**b**) three-dimensional image for the sensing silicon structure.

**Figure 6 sensors-21-06428-f006:**
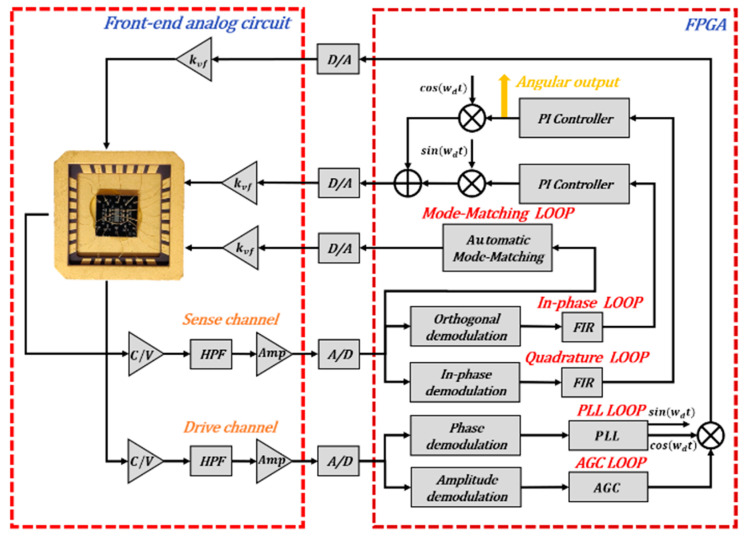
A schematic depiction of the control circuit system for dual-mass TFGs.

**Figure 7 sensors-21-06428-f007:**
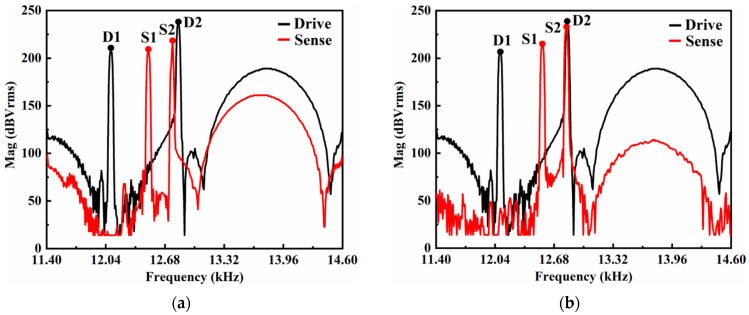
Comparisons of resonance frequencies (**a**) before and (**b**) after automatic mode-matching of dual-mass TFGs.

**Figure 8 sensors-21-06428-f008:**
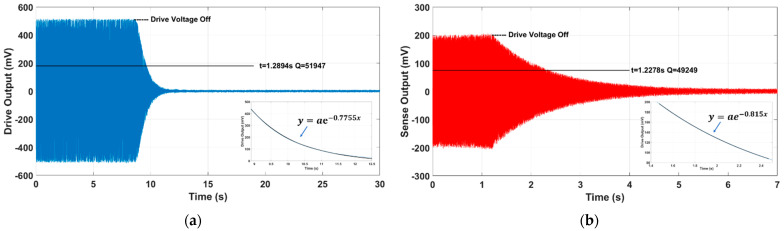
The time domain dynamic testing results for (**a**) the drive mode and (**b**) the sense mode of dual−mass TFGs.

**Figure 9 sensors-21-06428-f009:**
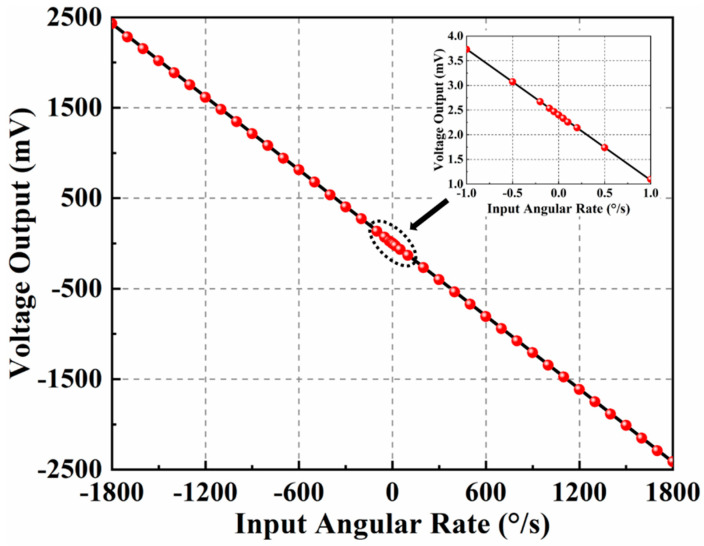
Full−scale testing results and nonlinearity of TFGs after automatic mode−matching.

**Figure 10 sensors-21-06428-f010:**
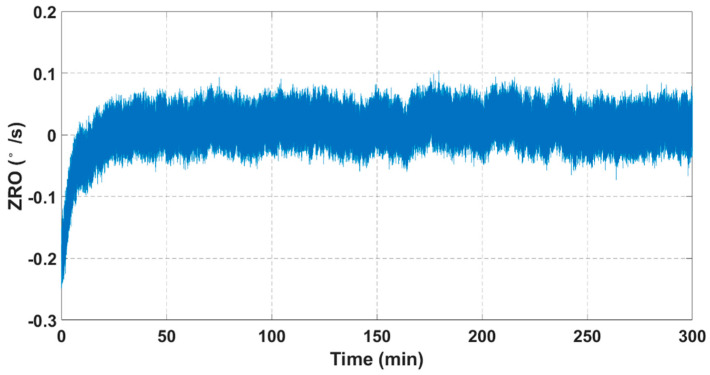
Room−temperature ZRO results.

**Figure 11 sensors-21-06428-f011:**
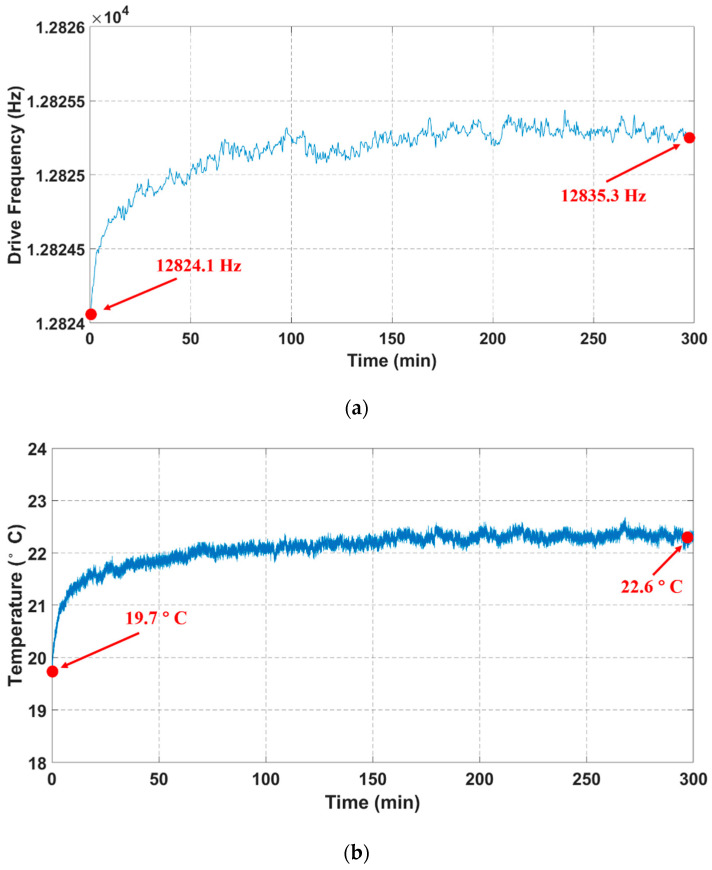
Synchronous sampling results for: (**a**) drive frequencies and (**b**) device temperatures of TFGs.

**Figure 12 sensors-21-06428-f012:**
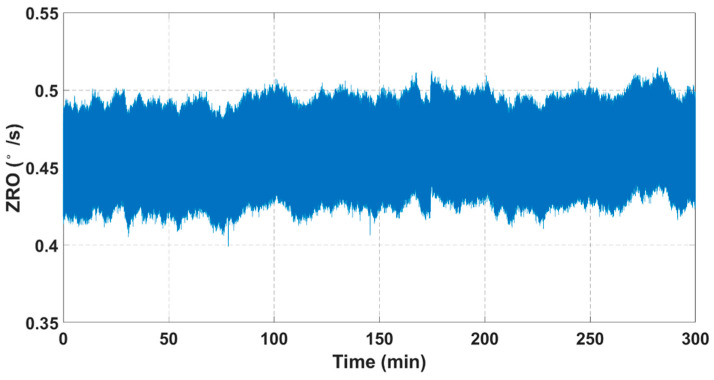
Fitted ZRO results suffering a third-order temperature compensation.

**Figure 13 sensors-21-06428-f013:**
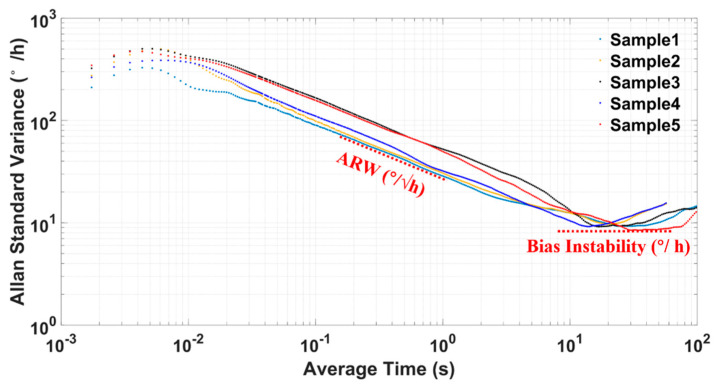
The Allan standard variance analysis results for typical TFGs.

**Table 1 sensors-21-06428-t001:** Key parameters of the dual-mass TFGs.

Parameters	Values
Structure thickness	100 μm
Proof mass	0.87 mg
Drive capacitance	1.85 pF
Sense capacitance	3.25 pF
Electrode length	150 μm
Electrode width	6 μm
Drive frequency	12,824 Hz
Sense frequency	12,768 Hz
Q-factor of drive mode	51,947
Q-factor of sense mode	49,249
Full-scale range	±1800°/s
Scale factor Nonlinearity	1.345 mV/(°/s) 720 ppm/°C

**Table 2 sensors-21-06428-t002:** Performance comparison results of five typical TFGs.

Samples	Angle Random Walk (°/√h)	Bias Instability (°/h)
Sample1	0.947	9.277
Sample2	0.976	9.699
Sample3	0.891	9.175
Sample4	0.883	9.185
Sample5	0.918	9.015

## Data Availability

Not applicable.

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
