# Peer review of "A Novel High-Q Dual-Mass MEMS Tuning Fork Gyroscope Based on 3D Wafer-Level Packaging"

_sensors, 2021, doi:10.3390/s21196428_

Round 1

Reviewer 1 Report

This paper describes the development of a gyroscope using 3D wafer-level packaging. I believe it is worth to be published. Some amendments are recommended to improve readability of the paper.

  1. The authors state that the resolution of the gyroscope is 0.1°/s according to figure 9. Can authors give some more details for the measurement, since I didn’t get any information of resolution in figure 9.
  2. The authors state that the Zr-based getter is used for maintain the vacuum in the chamber. To the best of my knowledge, the vacuum is very important for TFGs. Can authors clarify that the hermetic seal and the vacuum packaging results for the measured devices?

Reviewer 2 Report

This paper is untitled A Novel High Q Dual Mass MEMS Tuning Fork Gyroscope Based On 3D Wafer Level Packaging. It is about authors proposes, and they experimentally demonstrates, a news high quality factor dual mass tuning fork Micro Electro Mechanical System (MEMS) gyroscope utilizing 3-dimensional packaging techniques. Authors annouce that Q factors of the drive and sense modes reach up to respectively 51947 and 49249. We remind that the displacement of tuning fork gyroscope (TFG ) relatively to the plane of oscillation, is measured to produce a signal related to the rotational speed of the whole system. According to the authors of this manuscript, their TFG realizes a wide measurement range of ± 1800 ° /s and a high resolution of 0.1 ° / s with a scale-factor nonlinearity 720 ppm after automatic mode-matching. Long term Zero Rate Output (ZRO) drift can be effectively suppressed by temperature compensation. It induces a small angle random walk (ARW) of 0.923 ° / √h and also a low bias instability (BI) of 9.270 ° / h.

I now go into the review report more specifically. I want to say that I find the summary is perfectly relevant. Everything is there. I think there is nothing to change there. Same remark on the key words which give an account of the subject treated in the proposed article.

I am also quite enthusiastic about the introduction which I also find interesting and suitable for what is needed for this article. References are correctly cited.

Now some remarks and questions:

1/ Minor: You should indicate in a little bit bigger caracters the words indicated in red in Figure 1. A schematic illustration of the dual mass TFG sensing element, page 3, line 103.

2/ Minor: Page 11. Line 327 – 337: Delete the two paragraphs concerning Appendices A and B.

For remark 3/ to 6/ it is better to add the digital object identifier (DOI) link:

3/ Minor: Page 12. Reference 6. Please cite the corresponding DOI of this conference paper: DOI: 10.1109/PLANS.2006.1650648 https://doi.org/10.1109/PLANS.2006.1650648

4/ Minor: Page 12. Reference 9. Please cite the corresponding DOI of this conference paper: DOI: 10.1109/SENSOR.2009.5285411 https://doi.org/10.1109/SENSOR.2009.5285411

5/ Minor: Page 12. Reference 18. Please cite the corresponding DOI of this conference paper: DOI: 10.1109/MEMSYS.2007.4433009 https://doi.org/10.1109/MEMSYS.2007.4433009

6/ Minor: Page 12. Reference 20. Please cite the corresponding DOI of this conference paper: DOI: 10.1109/ISISS.2014.6782536 https://doi.org/10.1109/ISISS.2014.6782536

It is therefore time to conclude this review. I would like to congratulate the authors for their work and for this proposed manuscript, and if there were not some little improvements to the margin, like for their reference list DOI references of some conference papers, I find that this paper should be logically accepted for publication in the journal. I would say with minor revision so that authors can marginally improve their manuscript.

Reviewer 3 Report

In this paper Authors proposes and experimentally demonstrates a novel high-Q dual mass tuning fork microelectromechanical system (MEMS) gyroscope utilizing three-dimensional (3D) packaging techniques.

Nice paper presented MEMS from idea trough design, simulation, fabrication and characterization.

From cosmetic point of view - Figure 1, 2, 3 are too small.
